# Association between Health-Related Quality of Life and Completion of First-Line Treatment among Lung Cancer Patients

**DOI:** 10.3390/cancers14143343

**Published:** 2022-07-09

**Authors:** Anne Katrine G. Levinsen, Susanne O. Dalton, Ingelise Andersen, Anders Mellemgaard, Marianne S. Oksen, Lena Saltbæk, Nina H. G. Hansen, Signe Carlsen, Trille K. Kjaer

**Affiliations:** 1Survivorship and Inequality in Cancer, Danish Cancer Society Research Center, 49 Strandboulevarden, 2100 Copenhagen, Denmark; sanne@cancer.dk (S.O.D.); lsal@regionsjaelland.dk (L.S.); ninagantzel@gmail.com (N.H.G.H.); signec92@hotmail.com (S.C.); trille@cancer.dk (T.K.K.); 2Danish Research Center for Equality in Cancer, Department of Clinical Oncology & Palliative Care, Zealand University Hospital, Rådmandsengen 5, 4700 Næstved, Denmark; 3Department of Public Health, University of Copenhagen, Øster Farimagsgade 5, 1014 Copenhagen, Denmark; inan@sund.ku.dk; 4Department of Oncology, Herlev University Hospital, Borgmester Ib Juuls Vej 1, 2730 Herlev, Denmark; a.mellemgaard@dadlnet.dk (A.M.); marianne.stensoee.oksen@regionh.dk (M.S.O.)

**Keywords:** quality of life, cancer care, lung cancer

## Abstract

**Simple Summary:**

The aim of this study was to investigate how health-related quality of life at time of diagnosis is associated with the completion of planned first-line oncological treatment among lung cancer patients. Patients with reduced function and patients who reported fatigue, pain, appetite loss, and financial difficulties at time of diagnosis had significantly increased adjusted odds ratios for not completing the planned first-line oncological treatment. Measures of lung cancer patients’ self-reported HRQOL as part of the diagnostic evaluation at time of diagnosis may contribute to the optimization of planned oncological treatment.

**Abstract:**

Experts recommend assessing lung cancer patients’ health-related quality of life (HRQOL) in the diagnostic evaluation. We investigated the association between HRQOL and completion of first-line treatment among lung cancer patients in a prospective cohort study. Clinical information on lung cancer patients was obtained from medical records, and information on quality of life and lung cancer-related symptoms was obtained through questionnaires at time of diagnosis. We used directed acyclic graphs to identify potential confounders and mediators between HRQOL and completion of first-line treatment. The association between functioning levels and symptoms and completion of first-line oncological treatment was estimated as odds ratios, with 95% confidence intervals, in logistic regression models. In all, 137 patients (52% men, mean age: 69 years) participated, out of 216 invited. Patients who reported reduced functioning had significantly increased ORs for not completing first-line treatment: poor physical function (OR 4.44), role function (OR 6.09), emotional function (OR 5.86), and social function (OR 3.13). Patients with fatigue (OR 7.55), pain (OR 6.07), appetite loss (OR 4.66), and financial difficulties (OR 17.23) had significantly increased ORs for not completing the first-line treatment. Reduced functioning and presence of symptoms were associated with not completing first-line treatment. An assessment of HRQOL could potentially aid the diagnostic evaluation and treatment planning for lung cancer patients.

## 1. Introduction

Health-related quality of life (HRQOL) is an important prognostic factor in patients with cancer [1,2,3,4], including lung cancer patients [5,6,7,8]. Clinical experts recommend the assessment of HRQOL as an estimation of disease influence on lung cancer patients [9,10,11], which can be used in combination with performance status (PS), comorbidity, stage, and histopathology when planning cancer treatment [12].

A study by Movsas et al. [8] showed that self-reported HRQOL is an independent prognostic factor for survival among lung cancer patients receiving chemoradiotherapy, and that HRQOL might replace or supplement other prognostic factors, such as PS and stage at diagnosis. This indicates that patient-reported HRQOL might be just as relevant as a prognostic factor as PS assigned by health professionals [8]. Other studies show that the association between HRQOL and cancer survival may partly be explained by a psychosocial connection, in which poor HRQOL is associated with lack of adherence to treatment plans [7,13,14].

A nationwide Danish study found that 46% of lung cancer patients did not receive the recommended first-line treatment [12]. A few studies have shown that self-reported HRQOL is associated with planned, as well as actually received, treatment among cancer patients [15,16,17,18]. Furthermore, a study by Daroszewski et al. found that specific domains or parts of HRQOL (decreased physical functioning, loss of appetite, and dysphagia) reported prior to initiation of chemotherapy were associated with not completing the planned treatment among lung cancer patients [19]. No other studies have, to our knowledge, looked at the association between HRQOL and completion of oncological treatment among lung cancer patients. Thus, the aim of this prospective study was to investigate the association between HRQOL and completion of planned first-line oncological treatment among newly diagnosed patients with lung cancer.

## 2. Materials and Methods

### 2.1. Study Population

A total of 216 patients diagnosed with non-small-cell lung cancer (NSCLC) or small-cell lung cancer (SCLC) were invited to participate in the study at the time of their first contact with the oncology department at Herlev University Hospital, from September 2016 through to September 2017. After informed consent was obtained from participants, questionnaires were filled out at three different time points (baseline, first follow-up at end of treatment, second follow-up 1 month after end of treatment).

### 2.2. Quality of Life and Lung Cancer-Related Symptoms

Quality of Life was assessed by the European Organization for Research and Treatment of Cancer Quality of Life Core Questionnaire (EORTC QLQ-C30) [20] and the Lung Cancer Module (QLQ-LC13) [21], which has shown acceptable reliability and validity in psychometric testing among lung cancer patients [21,22]. The QLQ-C30 comprises one global quality of life scale, five functioning scales (physical, role, cognitive, emotional, and social), three symptom scales (fatigue, pain, and nausea and vomiting), and six single items (dyspnea, loss of appetite, insomnia, constipation, diarrhea, and financial impact of disease). The QLQ-LC13 comprises one multi-item scale (dyspnea) and ten single items (coughing, hemoptysis, sore mouth, dysphagia, peripheral neuropathy, alopecia, pain in chest, pain in arm or shoulder, and pain in other parts). Each item was rated on a four-point scale as “not at all”, “a little”, “quite a bit”, and “very much”, except for the global quality of life scale, which is rated on a seven-point scale from “very poor” to “excellent”. We calculated the scores for both scales and single items by linear transformation to a range of 0–100 according to published guidelines [23]. HRQOL will, in the following, be used as one concept that covers both functioning and lung cancer-related symptoms.

### 2.3. Completion of First-Line Treatment

Information on whether the planned first-line treatment was palliative or curatively intended, treatment complications, dose reductions, and dose delays were attained from patients’ medical records. First-line treatment was defined as the intended treatment with either immunotherapy, chemotherapy, or radiotherapy as stand-alone treatments, or in combination. We did not have information on individual-level mutation status, however, treatment according to national guidelines was practiced. Completion of first-line treatment was defined as a consistency between the intended treatment and the actual treatment received. Omission of ≥20% reductions in one or more chemotherapy doses, or a delay in radiotherapy, were classified as deviation from the planned treatment. Reasons for inconsistency between intended treatment and actual treatment received were toxicity, nonattendance, disease progression or lack of response to therapy, or death. Completion of first-line treatment was dichotomized into yes or no.

### 2.4. Covariates

Information on education, marital status, smoking habits, alcohol intake, and body mass index (BMI) of the lung cancer patients were self-reported, and obtained through the baseline questionnaire. Highest attained education was categorized into short (9 years or less), medium (10–12 years), and long education (>12 years). Marital status was categorized as being married, cohabiting, single, widower or separated/divorced, and afterwards dichotomized into cohabiting or living alone. Smoking was categorized as never smoked, former smoker, or current smoker. Alcohol intake was defined as intake during the last week, and categorized as no alcohol intake (0 units), alcohol intake within the recommended amount by the Danish health authorities (1–7/14 units for women/men), and alcohol intake higher than the recommended amount (≥8/15 units for women/men). BMI was categorized as underweight (<18.5), normal weight (18.5–24.9), overweight (25–29.9), and obese (>30).

We obtained information on stage of disease, PS, and comorbidity of the lung cancer patients from medical records. The tumor node metastasis (TNM) classification was used for patients with NSCLC, and categorized into early disease stage (surgically treated and T1-3, N0-1, M0), medium disease stage (locally advanced and T1-4, N0-3, M0), and advanced disease stage (any T, any N, M1). For SCLC, we used the terms limited (M0) and extended (M1) disease. Limited disease was categorized as early stage disease, while extended disease was categorized as advanced disease. Eastern Cooperative Oncology Group (ECOG) PS was defined in the clinic, and is based on a scale from 0–5, where a score of 0 indicates great health, 1 indicates restrictions in physical strenuous activities, 2 indicates capability of self-care but being unable to carry out work-based activities, 3 indicates capability of limited self-care or being confined to a bed or chair, 4 indicates complete disability, and a score of 5 indicates death [24]. We retrieved information about patients’ comorbidity from medical records. We used a pre-specified list of conditions based on a systematic review by Barnett et al. [25], which included more than 40 morbidities and other long-term disorders, such as hypertension, coronary heart disease, and thyroid disorders [25]. The list thus included morbidities that are not always thought of as definite diseases, such as alcohol problems and hearing loss, but which still have an impact on patients’ health [25]. Comorbidity was categorized as having 0, 1, or 2, or more comorbidities.

### 2.5. Statistical Analysis

Each HRQOL domain was dichotomized using a cut-off value of <66.67 on the functional scales and >33.34 on the symptom scales and single items, which have been shown to be clinically relevant thresholds [26] denoting reduced functioning and presence of clinically relevant symptoms, respectively. We did not dichotomize the global HRQOL as this is not considered a function or symptom [23]. The association between functioning levels and lung cancer-related symptoms and completion of first-line treatment were estimated as Odds Ratios (ORs) with 95% confidence Intervals (CIs) in logistic regression models.

We used directed acyclic graphs (DAGs) [27,28] to identify potential confounders and mediators between HRQOL and completion of first-line treatment. DAG is a tool, which helps to graphically represent the causal mechanisms we assume are between exposure and outcome. Four models were performed; the first model was crude. In the second model we adjusted for sociodemographic factors (sex, age). In the third model we further included adjustments for comorbidity, disease stage, marital status, education, and lifestyle factors (smoking, alcohol intake, BMI). The final (fourth) model was further adjusted for PS, which was identified as a mediator between HRQOL and completion of first-line treatment in the DAG. If the estimated effect of the exposure (HRQOL) on the outcome (completion of first-line treatment) decreased after adjustment for PS, we interpreted PS as a partial mediator between HRQOL and completion of first-line treatment [29]. To handle missing data, we used list-wise deletion [30,31]. Statistical analysis were performed using the statistical software SAS Enterprise Guide 5.1 for Windows.

## 3. Results

Of the 216 patients who were invited to participate in the study, 63 patients declined and 10 patients only filled in the baseline questionnaire. This led to the inclusion of 143 newly referred lung cancer patients, with 137 responses available for analysis (Figure 1).

The mean age was 69 years, 52% were men, most were married (62%), previous smokers (73%), had a long education (57%), were diagnosed with NSCLC (85%), had a PS of 0 (53%), had two or more comorbidities (43%) and most did complete first-line oncological treatment (54%) (Table 1). Reasons for not completing first-line treatment were disease progression (5%), not meeting up for treatment (3%), death (5%) and toxicological reaction to treatment (75%). Most patients who completed first-line treatment received chemoradiotherapy (53%), while most patients who did not complete first-line treatment received chemotherapy only (51%). There was little difference between completing treatment and receiving radiotherapy only, or immunotherapy (Table 1).

Patients who did not complete first-line treatment reported lower mean functional levels for all functional scales, and higher mean levels of lung cancer-related symptoms for all items, except coughing, when compared to patients who did complete first-line treatment. The differences were small, however, for diarrhea, financial difficulties, sore mouth, peripheral neuropathy, and alopecia (Table 2).

The multivariate adjusted ORs for not completing first-line treatment were 3–6-fold increased among individuals with reduced functioning (physical function, OR 4.44, 95% CI: 1.52–14.32; role function, OR 6.09, 95% CI: 2.03–20.93; emotional function OR 5.86, 95% CI: 2.01–19.23; cognitive function, OR 3.06, 95% CI: 0.96–10.54; social function, OR 3.06, 95% CI: 0.96–10.54) compared to patients with good/normal functioning; however, the estimate for cognitive functioning did not reach statistical significance (Table 3, model III). The ORs were significantly increased for not completing first-line treatment among patients who reported the presence of one the following assessed symptoms: fatigue (OR 7.55, 95% CI: 2.44–27.42), pain (OR 8.24, 95% CI: 2.05–41.92), appetite loss (OR 4.66, 95% CI: 1.17–20.95), financial difficulties (OR 117.23, 95% CI: 1.12–587.1), and pain in other parts (OR 5.37, 95% CI: 1.46–22.74) (Table 3, model III).

Further adjustment for PS changed most of the ORs slightly towards the null on all functioning scales; however, ORs remained 2.5–5-fold higher among patients with reduced functioning, which was statistically significant for all functioning scales except for cognitive and social functioning (Table 3, model IV). For lung cancer-related symptoms, the ORs only remained statistically significant for fatigue, pain, and pain in other parts after further adjustment for PS, with increased ORs for not completing first-line treatment among patients reporting moderate–high levels of fatigue (OR 8.11 95% CI: 2.17–36.78) (Table 3, model IV). After adjustment for PS, we also found a significant association between experiencing alopecia and completion of treatment (OR 13.18 95% CI: 1.16–235.18), but the CIs were wide (Table 3, model IV).

## 4. Discussion

In this prospective study, we found that lung cancer patients with reduced physical, role, emotional, and social functioning and presence of fatigue, pain, appetite loss, financial difficulties, and pain in other parts of the body prior to the initiation of treatment were less likely to complete first-line treatment even after adjustment for relevant confounders. The results also indicate that PS mediates part, but not all, of the association between pretreatment functioning and symptoms and completion of treatment.

These findings align with previous studies on lung cancer patients, reporting that patients who did not complete chemotherapy had poorer physical functioning at time of diagnosis [19], and that poor HRQOL might be an indicator of noncompliance with the treatment plan [7]. In a systematic review, including 104 studies on quality of life and cancer survival, the authors suggested that HRQOL assessed before treatment might provide the most reliable information for assisting clinicians to establish prognostic criteria for treating cancer patients [15]. It is likely that patients who report poor HRQOL do not complete first-line treatment because these self-reported outcomes are markers of disease progression not picked up by the assignment of PS. Even though noncompletion of first-line treatment is affected by other factors as well [32], such as lack of response to therapy, it may be relevant to include an indicator for HRQOL when planning treatment among lung cancer patients.

In light of the relatively high rate of treatment noncompletion among lung cancer patients, one may wonder if HRQOL could replace or supplement PS as a measurement of overall health status. We found indications that part, but not all, of the associations between functioning and lung cancer-related symptoms and completion of first-line treatment was mediated by PS. A study by Movsas et al. found that HRQOL could supplement, or even replace, prognostic factors such as PS among NSCLC patients [8]. As PS is the medical overall assessment of the patients’ health status, and HRQOL is the patients’ own assessment of their health status, we argue that it may be valuable to also include measures of HRQOL when planning patients’ treatment. HRQOL is related to the individuals health, and systematically integrating information about cancer patients’ HRQOL may contribute to a more realistic treatment plan for the patients [15]. This is supported by findings from a study by Montazeri et al., where PS did not predict survival, whereas initial HRQOL was a significant predictor of survival after lung cancer [33].

HRQOL might contribute to the clinical evaluation of lung cancer patients at time of diagnosis for several reasons [3]. Firstly, HRQOL includes symptoms and domains that are not included in standard clinical measures, such as emotional function, and HRQOL also provides options for multiple responses. Secondly, HRQOL has the potential to pick up relevant information earlier, as studies have shown that changes in HRQOL can be an early warning of disease progression [3]. Thereby, HRQOL could have the potential to add valuable information when a treatment plan is decided between the patient and the clinician. Furthermore, HRQOL could provide knowledge on pretreatment symptoms that may be managed to enable the patient to complete treatment, and initiatives to improve HRQOL before treatment initiation could enhance chances for completing first-line treatment. Along the same line, baseline HRQOL could be used as stratifying variables for patients enrolled in randomized studies, to warrant a balanced distribution between study arms.

This study has several strengths. The prospective collection of data ensured temporal separation between exposure and outcome and minimal selection bias. Other strengths are the inclusion of data from both medical records and questionnaires, the use of validated scales and listwise deletion [30], as well as the use of DAGs to identify potential confounders and mediators [27,34,35]. To provide an indication as to what degree PS mediates the effect of HRQOL on completion of first-line treatment, we performed a mediation analysis, as proposed by Baron & Kenny [36].

Our study also has limitations. There is a risk of misclassification, as we dichotomized the HRQOL measurements. This was, however, necessary, as we were interested in the difference in completion of treatment among patients reporting reduced levels of functioning and having lung cancer-related symptoms at a clinically relevant level. Concerning the generalization of this study, we expect that more patients with better HRQOL participated in the study compared to the overall population of lung cancer patients referred for oncological treatment. This may have affected the absolute proportions of patients scoring low versus high on HRQOL measurements, but should, however, not affect the relative associations. A study by Dalton et al. (2015) found that, among a nationwide cohort of lung cancer patients, 46% did not receive the recommended first-line treatment [12]. This finding corresponds well with the finding in this study, where a similar proportion of the study population did not complete first-line treatment.

## 5. Conclusions

Lung cancer patients’ with self-reported HRQOL measures of reduced physical, role, emotional, and social functioning, and presence of lung cancer-related fatigue, pain, appetite loss, financial difficulties, and other pain types were associated with not completing first-line treatment. Adding measures of lung cancer patients’ HRQOL to the diagnostic evaluation may contribute to the optimization of planned oncological treatment.

## Figures and Tables

**Figure 1 cancers-14-03343-f001:**
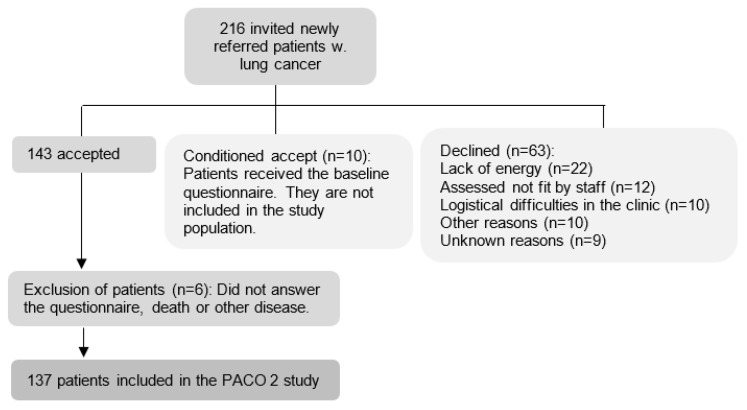
Flowchart for inclusion and exclusion of patients, Herlev University Hospital, Denmark, 2016–2017.

**Table 1 cancers-14-03343-t001:** Study population characteristics of 137 newly referred lung cancer patients in the PACO2 study, Herlev University Hospital, Denmark, 2016–2017.

Variable		Completed First-Line Treatment **		
	All n (%)	Yes n (%)	No n (%)	*p*-Value
Sex				
Men	71 (52)	40 (54)	31 (49)	0.6
Women	66 (48)	34 (46)	32 (51)	
Age at diagnosis (years)				
≤65	35 (26)	21 (28)	14 (22)	0.5
>65 to <70	41 (30)	24 (32)	17 (27)	
≥70 to <75	39 (28)	20 (27)	19 (30)	
≥75	22 (16)	9 (12)	13 (21)	
Mean (SD)	68.9 (7.4)	67.9 (7.8)	70.1 (7.9)	0.1
Marital status				
Cohabiting	96 (70)	52 (70)	44 (70)	1.0
Living alone	41 (30)	22 (30)	19 (30)	
Smoking status				
Never/used to smoke	107 (79)	55 (74)	52 (85)	0.2
Current smoker	28 (21)	19 (26)	9 (15)	
Alcohol				
0 units per week	47 (35)	26 (36)	21 (35)	0.1
>1 to <7/14 units per week	59 (44)	28 (38)	31 (52)	
>8/15 units per week	27 (20)	19 (26)	8 (13)	
BMI				
Underweight (<18.5)	11 (8)	-	-	0.004 *
Normal weight (18.5–24.9)	60 (44)	41 (55)	19 (31)	
Overweight (25–29.9)	45 (33)	15 (20)	30 (49)	
Obese (≤30)	19 (14)	11 (15)	8 (13)	
Education				
Short	26 (19)	13 (18)	13 (21)	0.7
Medium	23 (17)	14 (19)	9 (14)	
Long	78 (57)	40 (55)	38 (60)	
Other	9 (7)	-	-	
Performance score				
0	71 (53)	44 (59)	27 (44)	0.04 *
1	54 (40)	28 (38)	26 (43)	
2	10 (7)	-	-	
Diagnosis				0.3
NSCLC	116 (85)	65 (88)	51 (81)	
SCLC	21 (15)	9 (12)	12 (19)	
Stage				
Early/medium	75 (55)	41 (55)	33 (53)	0.5
Advanced	62 (45)	33 (45)	30 (47)	
Comorbidity				
0	34 (25)	24 (32)	10 (16)	0.08
1	44 (32)	22 (30)	22 (35)	
≥2	59 (43)	28 (38)	31 (49)	
Surgery before oncological treatment				
Yes	23 (17)	11 (15)	12 (19)	0.5
No	114 (83)	63 (85)	51 (81)	
Planned treatment				
Chemotherapy	59 (43)	27 (36)	32 (51)	0.2
Radiotherapy Chemoradiotherapy	4 (3) 63 (46)	3 (4) 39 (53)	1 (2) 24 (38)	
Immunotherapy	11 (8)	5 (7)	6 (9)	

* Significant at the 5% level. ** Consistency between the intended treatment and the actual treatment received. Due to low numbers (>5) few of the categories are collapsed. Missing values are not reported.

**Table 2 cancers-14-03343-t002:** Pretreatment quality of life and symptom burden by completion of first-line treatment in 137 newly referred patients with lung cancer, Herlev University Hospital, Denmark, 2016–17.

			Completed First-Line Treatment		
	N	Total Mean (SD)	Yes Mean (SD)	No Mean (SD)	*p*-Value
Functioning scales (QLQ-C30)					
Global QOL	135	62.1 (22.9)	68.2 (20.04)	55.2 (24.1)	<0.001 *
Physical function	137	77.3 (20.0)	83.6 (15.7)	69.9 (22.1)	<0.001 *
Role function	134	69.5 (27.99)	76.6 (25.7)	61.3 (27.4)	0.001 *
Emotional function	135	72.9 (22.4)	78.2 (19.1)	66.7 (24.5)	0.004 *
Cognitive function	135	87.7 (19.4)	90.9 (14.6)	83.6 (23.5)	0.035 *
Social function	134	86.9 (18.1)	92.0 (13.9)	80.9 (20.6)	<0.001 *
Symptom scales and single-items (QLQ-C30 and QLQ-LC13)					
Fatigue	136	34.1 (23.6)	28.3 (20.3)	41.0 (25.4)	0.002 *
Nausea/vomiting	136	7.23 (15.8)	4.7 (13.7)	10.2 (17.7)	0.049 *
Pain	135	16.7 (21.8)	12.2 (16.6)	22.1 (25.9)	0.011 *
Dyspnea	134	36.3 (30.2)	30.6 (27.6)	43.2 (31.8)	0.016 *
Insomnia	136	28.2 (31.92)	23.9 (29.5)	33.3 (34.1)	0.085
Appetite loss	136	25.98 (31.9)	20.3 (29.6)	32.8 (33.3)	0.02 *
Constipation	134	7.5 (17.6)	6.3 (14.3)	8.9 (21.1)	0.42
Diarrhea	134	7.7 (17.3)	7.3 (16.9)	8.2 (17.9)	0.77
Financial difficulties	135	5.4 (16.9)	4.5 (14.9)	6.6 (19.1)	0.49
Dyspnea (lung cancer-specific)	137	27.4 (24.6)	22.8 (22.2)	32.7 (26.9)	0.02 *
Coughing	137	43.6 (29.9)	45.5 (29.5)	41.3 (30.4)	0.41
Hemoptysis	137	6.3 (16.4)	4.9 (14.3)	7.9 (18.7)	0.30
Sore mouth	137	3.6 (11.2)	3.6 (11.8)	3.7 (10.6)	0.96
Dysphagia	136	6.4 (16.5)	5.5 (17.6)	7.4 (15.2)	0.50
Peripheral neuropathy	136	5.6 (14.9)	4.95 (11.9)	5.5 (17.6)	0.58
Alopecia	134	4.2 (17.5)	4.1 (17.4)	4.4 (17.9)	0.90
Pain in chest	134	14.9 (22.2)	10.9 (19.3)	19.7 (24.6)	0.027 *
Pain in arm or shoulder	134	12.4 (22.6)	11.7 (22.4)	13.3 (23.1)	0.68
Pain in other parts	132	18.4 (27.7)	13.9 (24.1)	24.1 (31.1)	0.042 *

* Significant at the 5% level.

**Table 3 cancers-14-03343-t003:** Logistic regression for the association between pretreatment quality of life and not completing first-line treatment among 137 lung patients, Herlev University Hospital, Denmark, 2016–17.

	Model I	Model II	Model III	Model IV
	OR (95% CI)	OR (95% CI)	OR (95% CI)	OR (95% CI)
Functioning Scales (EORTC QLQ-C30) Score <66.67 (Ref = High Function)				
Physical functioning	3.15 (1.5–6.82)	2.92 (1.38–6.38)	4.44 (1.52–14.32)	4.25 (1.37–14.56)
Role functioning	3.31 (1.62–6.97)	3.39 (1.64–7.22)	6.09 (2.03–20.93)	5.47 (1.69–20.82)
Emotional functioning	3.40 (1.67–7.09)	3.48 (1.68–7.44)	5.86 (2.01–19.23)	5.48 (1.78–18.95)
Cognitive functioning	1.84 (0.80–4.34)	1.76 (0.76–4.19)	3.06 (0.96–10.54)	3.31 (0.95–12.61)
Social functioning	2.87 (1.30–6.61)	2.93 (1.31–6.84)	3.13 (1.01–10.46)	2.73 (0.84–9.49)
Symptom scales and items (EORTC QLQ-C30 and QLQ-LC13 ^a^) score >33.33 (ref = no symptoms)				
Fatigue	4.02 (1.90–8.85)	3.97 (1.84–8.88)	7.55 (2.44–27.42)	8.11 (2.17–36.78)
Nausea/vomiting ^a^	6.40 (1.00–124)	6.44 (0.99–126)	6.07 (0.46–174.2)	5.40 (0.36–164.93)
Pain	5.71 (1.93–21.00)	5.93 (1.93–22.16)	8.24 (2.05–41.92)	7.91 (1.87–41.28)
Dyspnea	2.21 (1.02–4.94)	2.10 (0.95–4.73)	2.17 (0.72–6.92)	1.92 (0.57–6.68)
Insomnia	2.23 (1.01–5.08)	2.31 (1.02–5.38)	2.41 (0.79–7.75)	2.01 (0.62–6.75)
Appetite loss	2.62 (1.12–6.41)	2.95 (1.23–7.50)	4.66 (1.17–20.95)	3.85 (0.90–18.58)
Constipation	2.52 (0.24–54.91)	2.63 (0.24–57.91)	10.13 (0.23–543.6)	11.18 (0.18–882.28)
Diarrhea	3.72 (0.46–76.36)	4.49 (0.53–96.00)	7.32 (0.35–334.5)	8.51 (0.39–460.26)
Financial difficulties	3.78 (0.47–77.41)	4.77 (0.58–99.56)	17.23 (1.12–587.1)	11.56 (0.65–444.23)
Dyspnea (lung cancer-specific)	2.35 (1.07–5.31)	2.11 (0.94–4.89)	2.69 (0.88–8.86)	2.26 (0.65–8.2)
Coughing	0.89 (0.44–1.78)	0.95 (0.47–1.92)	0.81 (0.29–2.19)	0.73 (0.24–2.09)
Hemoptysis	1.18 (0.14–10.07)	1.37 (0.16–12.04)	0.08 (0.002–2.24)	0.12 (0.003–2.85)
Dysphagia	0.38 (0.02–3.02)	0.41 (0.02–3.30)	0.95 (0.04–10.55)	1.01 (0.04–12.15)
Alopecia	1.90 (0.30–14.75)	1.73 (0.27–13.65)	7.77 (0.76–114.5)	13.18 (1.16–235.18)
Pain in chest	2.99 (0.92–11.51)	3.35 (1.00–13.41)	4.8 (0.95–22.26)	4.17 (0.84–24.71)
Pain in arm or shoulder	1.03 (0.28–3.60)	1.15 (0.31–4.09)	2.13 (0.46–10.10)	2.06 (0.41–10.58)
Pain in other parts	2.38 (0.93–6.46)	2.57 (0.98–7.15)	5.37 (1.46–22.74)	4.58 (1.12–21.08)

^a^ Sore mouth and peripheral neuropathy were not included for analysis due to limited data. Model I: crude. Model II: adjusted for sex and age. Model III: adjusted for sex, age, education, marital status, smoking, alcohol, BMI, stage, comorbidity, treatment. Model IV: adjusted for sex, age, education, marital status, smoking, alcohol, BMI, stage, comorbidity, treatment, and performance status.

## Data Availability

The supporting data are not publicly available due to research participant privacy restrictions.

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
