# Peer review of "Association between Health-Related Quality of Life and Completion of First-Line Treatment among Lung Cancer Patients"

_cancers, 2022, doi:10.3390/cancers14143343_

Round 1

Reviewer 1 Report

The authors performed a prospective study to evaluate the prognostic role of QoL on compliance of treatment in the setting of lung cancer. The research question is interesting and the methodology seems valid. 

I suggest to better explain the DAGs methodology in the methods section and underline the reason for the choice of such method, becaus a medium reader might be not familiar with it

As for the result I wonder why Global QOL (that is the QoL item most used to summarize the QoL status) is out of the logistic regression. I suggest to include it. Also in the light of the fact that for using this evidence to characterize a patient as (good/poor prognosis for compliance), you need a summarizing dicotomic variable.

As for results interpretation, I suggest to also discuss the role of baseline QoL as prognostic factor to be used for stratifying (in place of PS) the population enrolled in randomised studies, to warrant a balanced distribution between study arms.

Author Response

Reviewer 1

The authors performed a prospective study to evaluate the prognostic role of QoL on compliance of treatment in the setting of lung cancer. The research question is interesting and the methodology seems valid. 

Comment:

I suggest to better explain the DAGs methodology in the methods section and underline the reason for the choice of such method, because a medium reader might be not familiar with it

Authors reply:

I have shortly explained the use of DAGs in the methods section page 3-4, line 144-146. DAGs is a tool, which graphically represents the causal mechanisms that we assume are at play between the exposure and outcome.  

Comment:

As for the result I wonder why Global QOL (that is the QoL item most used to summarize the QoL status) is out of the logistic regression. I suggest to include it. Also in the light of the fact that for using this evidence to characterize a patient as (good/poor prognosis for compliance), you need a summarizing dicotomic variable.

Authors reply:

This is a indeed a valid point. We did not calculate severity scores for Global QOL as this is not considered a symptom or function. Global QOL is scored differently from the other functions and symptoms included in EORTC QLQ-C30 (Fayers et al., 2001). Global QOL is scored from 1-7 (“very poor” to “excellent”) and even though it is possible to identify those who report either very poor or excellent self-reported global QOL, no cut-off point for this items exist. This is now clarified in the manuscript page 3, line 138-139.

Comment:

As for results interpretation, I suggest to also discuss the role of baseline QoL as prognostic factor to be used for stratifying (in place of PS) the population enrolled in randomised studies, to warrant a balanced distribution between study arms.

Authors reply:

This is an excellent point, which we have added to the Discussion section page 9, line 255-259.

Reviewer 2 Report

This earnest study could be improved at the following advices.

1. The significance of the study could be advocation to improve the HRQOL for enhancing completion chance of planned first-line treatment.

2. Please well define the performance status (PA) and its scoring for the readers to understand.

3. The numbers of included patients on Line 150 is different from that of Figure 1.

4.  On Line 168, "...Due to low numbers (>5) ...", it could be (<5).

5. The data were presented plainly. The discussion is too simple to attract reader's mind. Please address the importance of your findings and potential application for clinical approach in the future. 

Author Response

Reviewer 2

This earnest study could be improved at the following advices.

Comment:

  1. The significance of the study could be advocation to improve the HRQOL for enhancing completion chance of planned first-line treatment.

Authors reply:

We agree that the results of this study should raise attention to the potential benefit of improving HRQOL before treatment initiation to optimize chances for completing planned first-line treatment. We have now added this aspect to the Discussion section, page 9, line 255-259.

Comment:

  1. Please well define the performance status (PA) and its scoring for the readers to understand.

Authors reply:

We have now clarified further the scoring of PS, page 3, line 122-125.

Comment:

  1. The numbers of included patients on Line 150 is different from that of Figure 1.

Authors reply:

We are very sorry for the confusion and have now corrected the number.

Comment:

  1. On Line 168, "...Due to low numbers (>5) ...", it could be (<5).

Authors reply:

Thank you for pointing this out. We have now corrected it.

Comment:

  1. The data were presented plainly. The discussion is too simple to attract reader's mind. Please address the importance of your findings and potential application for clinical approach in the future. 

Authors reply:

We thank you for this comment. In reply to various specific comments from this and the other Reviewers we have elaborated on the importance of our findings and potential implications hereof (page 9, line 255-259). This include the potential for improving compliance by addressing HRQOL at diagnosis and on relevant use of HRQOL as a stratifying variable in RTCs instead of or in combination with PS to ensure better balancing of study arms.

Reviewer 3 Report

It is interesting

I'd like to know what type of dyspnea scale is used

Please, could you provide specific information about staging and comorbidities?

What diagnostic technique was applied for determining the staging?

Was the depressive mood considered as an item conditioning the quality of life? please provide pack-years index.

What about the mutation state?It is needed a clarification about types of first line treatment

The significance level is not reported in table 2

Please include reference about age and chemotherapy choice

-Clin Transl Oncol. 2019 Jun;21(6):790-795

Author Response

Reviewer 3

It is interesting

Comment:

I'd like to know what type of dyspnea scale is used

Authors reply:

Information on dyspnea was included as a single item as part of the validated and widely used EORTC-QLQ-C30. Question: ”Were you short of breath?”.
Information about lung cancer specific dyspnea as a symptom-scale is part of the validated EORTC QLQ-LC13. Questions included: “Were you short of breath when you rested?”, “Were you short of breath when you walked?”, “Were you short of breath when you climbed stairs?”

Comment:

Please, could you provide specific information about staging and comorbidities?

Authors reply:

For information on staging we kindly refer to page 3, line 115-119.

For information on comorbidity we have now added some examples of comorbid conditions in the text (page 3, line 129) and also present the comorbidities in a new Table (Supplementary table 1). Due to few numbers we however, cannot present numbers affected by each condition. Whether or not the extra table should be included in the final manuscript or as an additional online material may be an editorial decision.

Supplementary table 1. Conditions included as comorbid conditions

Hypertension

Depression

Painful condition

Asthma

Coronary heart disease

Treated Dyspepsia

Diabetes

Thyroid disorders

Rheumatoid arthritis. Other inflammatory polyarthropathies and systematic connective tissue disorders including arthrosis

Hearing loss

Chronic obstructive pulmonary disease/lung fibrosis

Anxiety and other neurotic, stress related and somatoform disorders

Irritable bowel syndrome

Other cancer

Alcohol problems

Other psychoactive substance misuse

Treated constipation

Stroke and transient ischemic attack

Chronic kidney disease

Diverticular disease of intestine

Arterial fibrillation/arrhythmia

Peripheral vascular disease

Heart failure

Prostate disorders

Glaucoma

Epilepsy (currently treated)

 Dementia

Schizophrenia ( and related non-organic psychosis) or bi-polar disorder

Psoriasis or eczema

Inflammatory bowel disease

Migraine

Blindness and low vision

Chronic sinusitis

Learning disability

Anorexia or bulimia

Bronchiectasis

 Parkinson's disease

 Multiple sclerosis

 Viral hepatitis

Chronic liver disease

Hypercholesterolemia

Periphery nerve disease

Chronic back pain

ADHD

Osteoporosis

Aortic stenosis, aorta insufficiency, aorta aneurism

Comment:

What diagnostic technique was applied for determining the staging?

Authors reply:

We are sorry for this but we did not have permission or opportunity to collect information on staging technique as staging was done in another department. We obtained information on stage from the medical records from the treating institution. However, national Danish guidelines for staging of lung cancer include recommendation to use an exact description of tumor spread and a CT or PET scan needs to include an exact indication of tumor size and localization, as well as an indication of which lymph nodes are pathological.

Comment:

Was the depressive mood considered as an item conditioning the quality of life? please provide pack-years index.

Authors reply:

A very good point. To some degree, depressive mood is part of the emotional functioning scale, which was also highly correlated to not completing first-line treatment.

Unfortunately, we did not have information on pack-years available only current smoking status

Comment:

What about the mutation state? It is needed a clarification about types of first line treatment

Authors reply:

Allocation to treatment according to mutation status followed national guidelines. Unfortunately, we did not have access to individual mutation status. Clinical experience from out co-authors was that guidelines were followed strictly. Thus we used the information on treatment plan, which should include treatment based on potential relevant mutation status. This is now clarified on page 3, line 94-95.

Comment:

The significance level is not reported in table 2

Authors reply:

This is now added to the table.

Comment:

Please include reference about age and chemotherapy choice: -Clin Transl Oncol. 2019 Jun;21(6):790-795

Authors reply:

This is now added to the reference list.

Round 2

Reviewer 1 Report

responses of the authors are satisfying